# Initial Treatment of High-Altitude Pulmonary Edema: Comparison of Oxygen and Auto-PEEP

**DOI:** 10.3390/ijerph192316185

**Published:** 2022-12-03

**Authors:** Markus Tannheimer, Raimund Lechner

**Affiliations:** 1Department of Sport and Rehabilitation Medicine, University of Ulm, Leimgrubenweg 14, 89075 Ulm, Germany; 2Department of General and Visceral Surgery, Krankenhaus Blaubeuren, Ulmer Straße 26, 89143 Blaubeuren, Germany; 3Department of Anesthesiology, Intensive Care, Emergency Medicine and Pain Therapy, Bundeswehr Hospital Ulm, Oberer Eselsberg 40, 89081 Ulm, Germany

**Keywords:** high-altitude, acute mountain sickness (AMS), Denali, Mount Everest, pursed lips breathing, expiratory positive airway pressure (EPAP)

## Abstract

Background: Improvement of oxygenation is the aim in the therapy of high-altitude pulmonary edema (HAPE). However, descent is often difficult and hyperbaric chambers, as well as bottled oxygen, are often not available. We compare Auto-PEEP (AP-Pat), a special kind of pursed lips breathing, against the application of bottled oxygen (O_2_-Pat) in two patients suffering from HAPE. Methods: We compare the effect of these two different therapies on oxygen saturation measured by pulse oximetry (SpO_2_) over time. Result: In both patients SpO_2_ increased significantly from 65–70% to 95%. Above 80% this increase was slower in AP-Pat compared with O_2_-Pat. Therapy started immediately in AP-Pat but was delayed in O_2_-Pat because of organizational and logistic reasons. Conclusions: The well-established therapies of HAPE are always the option of choice, if available, and should be started as soon as possible. The advantage of Auto-PEEP is its all-time availability. It improves SpO_2_ nearly as well as 3 L/min oxygen and furthermore has a positive effect on oxygenation lasting for approximately 120 min after stopping. Auto-PEEP treatment does not appear inferior to oxygen treatment, at least in this cross-case comparison. Its immediate application after diagnosis probably plays an important role here.

## 1. Introduction

High-altitude pulmonary edema (HAPE) is a life-threatening illness that may affect hitherto healthy persons after fast ascent to high altitudes [1]. It has a different pathophysiology compared with acute mountain sickness (AMS) and high-altitude cerebral edema (HACE) [2]. The aggravated increase in pulmonal arterial pressure (PAP) evoked by hypoxic-induced vasoconstriction (Euler-Liljestrand-Reflex) plays a major role [3,4]. This offers several therapeutic options with the aim of reducing PAP. Up to now slow release Nifedipine is the first line medication, Dexamethasone is effective too, whereas phosphodiesteraseinhibitors (Sildenafil, Tadalafil) have not fulfilled the expectations placed on them [5,6]. All medical treatment is off-label use as these pharmaceutical agents have no drug approval for the indication of HAPE. Although this is of secondary importance for emergency medical treatment by a physician, it is common practice for medical laypersons to use them. Due to legal problems regarding drug application according to western standards, this is of special relevance for medical laypersons.

As the primary underlying mechanism of HAPE is hypoxia the therapy of choice is to improve oxygenation. The general recommendation is to descend to the last asymptomatic sleeping altitude or at least 500 m of altitude. This is often difficult to realize because of organizational reasons (difficult terrain, infrastructure, bad weather, nighttime, and carrying/transportation options) or the patient’s condition. Another option to improve oxygenation is the use of a hyperbaric chamber or the use of bottled oxygen as the recommended gold standard [7,8,9]. Both must be carried along, of course. This is mostly the case for organized expedition mountaineering, but not for trekking, independent small expedition teams, remote regions, or alpine style with limited transport possibility. Therefore we want to present a simple and always applicable procedure, Auto-PEEP [10], and compare its effect with bottled oxygen in two patients suffering from HAPE.

## 2. Materials and Methods

We compare two patients with HAPE to demonstrate the effect of supplemental oxygen and Auto-PEEP on peripheral tissue oxygenation measured by pulse oximetry (SpO_2_). Auto-PEEP is a special breathing pattern we described earlier [10]. Controlled by a wristwatch 2 s of deep inspiration are followed by 8 s of expiration against the resistance of the pursed lips in a way that the exhaled air can be felt on the back of the hand at a distance of 30 cm. After the expeditions, we measured the pressure during this maneuver using intraoral piezo-resistive sensors (MIPM Mammendorf) [11] and could show that Auto-PEEP leads to an intraoral pressure of approximately 55 cm of H_2_O (min 40, max 65 cm H_2_O). As the vocal cords are open during Auto-PEEP we conclude that intrathoracic pressure is in the range of approximately 55 cm H_2_O.

In both cases we used a Nonin PalmSat 2500^®^ with a reusable flexible finger sensor (Adult Flexi-Form 9000A) placed on the index finger to visualize SpO_2_. This device logs the data for SpO_2_ and pulse every 4 s and has a capacity of 72 h [12]. Data were transferred to a laptop and analyzed after the expeditions.

The first patient (O_2_-Pat), male 55 years of age, a very experienced ultra-marathon runner, suffered from HAPE during a trek to Mera Peak (6461 m) located in the region of Mount Everest in Nepal. During the ascent to the high camp of Mera Peak on day 9 of the trek, at an altitude of about 5000 m and above, he lagged significantly behind the rest of the group. Up to this point he had been ascending without any problems and had always been in the upper performance level of the group. He complained of a sudden loss of performance and hallucinations. Since the terrain was easy, the wayfinding clear, and there was light for several more hours, he turned around shortly after the Mera La Pass at about 5450 m and descended alone to the last camp with permanent shelters at Karre (4870 m). The rest of the group climbed Mera Peak and returned to Karre the next morning. The health of the patient had subjectively improved somewhat in the meantime, with increasing appetite, but he felt very weak.

Later that evening, there was another sudden increase in respiratory distress. Self-measured SpO_2_ was between 40 and 50% as reported the following day. Self-medication with inhaled sympathomimetics and steroids was undertaken, but brought only a slight improvement. After a restless night, the patient’s condition had not improved the following morning so two physicians that were in Karre were consulted. The SpO_2_ with ambient air was 62% with a poor saturation signal, and the patient complained of marked dyspnea. The Lake Louise Score (LLS) [13] was 10 (incapacitating headache: 3; poor appetite: 1; incapacitating weakness: 3; mild dizziness: 1; AMS functional score: 2). HAPE was suspected and it was decided in agreement with the leader of the group that the patient should be transported to lower elevations as soon as possible. Emergency bottled oxygen was organized while transport was arranged. A portable hyperbaric bag was not available. In the meantime the patient was instructed in Auto-PEEP and tried to apply it, but he was too unsettled to follow the instructions correctly. Therefore, breathing cannot be called Auto-PEEP but pursed lips breathing (PLB). However, PLB could not be performed consistently because of exhaustion, indicated by increasing pulse during PLB. Therefore, PLB was stopped and luckily supplementary oxygen was available 5 min later. Therapeutically, until oxygen arrived, 8 mg of Dexamethasone, 20 mg of Nifedipine retard, and 800 mg of Ibuprofen were administered orally. He received bottled oxygen at an initial rate of 3 L/min for a short time, which was reduced to a constant rate of 0.5–1 L/min. Subsequently, it was possible to raise the SpO_2_ steadily from <70% to >95% (Figure 1).

The second patient (AP-Pat), male 33 years old, an experienced climber in the European Alps, tried to ascend Denali (6198 m) in Alaska. After his flight from Talkeetna (350 m) to the glacier (2100 m) he ascended to 4380 m within 4 days (1st day up to 2400 m, 2nd day up to 3300 m, 3rd day up to 4380 luggage transport, descent, night at 3300 m, 4th day up to 4380 m) carrying heavy loads. There he suffered from initial HAPE with significantly impaired physical performance and dyspnea and an LLS of 11 (incapacitating headache: 3; moderate nausea: 2; incapacitating weakness: 2; moderate dizziness: 2; functional score: 2). His SpO_2_ was 62% (59–65%), which is low compared with the findings of Roach et al. who reported an average SpO_2_ of 81.5 ± 4.4% in 102 healthy mountaineers in the same location [14]. At this altitude the advanced base camp (ABC), some also call it “Rangers Camp”, is located and radio communication is available. As symptoms occurred in the late afternoon the group stayed in this location with minimal infrastructure. AP-Pat performed Auto-PEEP by himself in a recumbent position with slightly elevated thorax (~15°) for 30 min followed by 2 h of normal breathing without any restrictions until Auto-PEEP started again. Both patient and accompanying physician observed SpO_2_ continuously on the display of the pulse oximeter. Nifedipine, Dexamethasone, and Acetazolamide were available but not used.

## 3. Results

After the administration of bottled oxygen SpO_2_ increased rapidly from ~64% to 97% within 2 min in O_2_-Pat (Figure 1). A solid hour after the start of the therapy (30 min after start of O_2_), he was evacuated to Lukla by helicopter for a first checkup. Several hours later the patient was transferred to a hospital in Kathmandu by helicopter. A physical examination delivered no pathological findings. In the chest X-Ray bilateral prominent bronchovascular markings were diagnosed, with blood and urine samples showing no abnormalities. The patient’s health improved rapidly and by the evening of the same day the patient felt subjectively almost healthy. He was discharged 3 days later in good health.

**Figure 1 ijerph-19-16185-f001:**
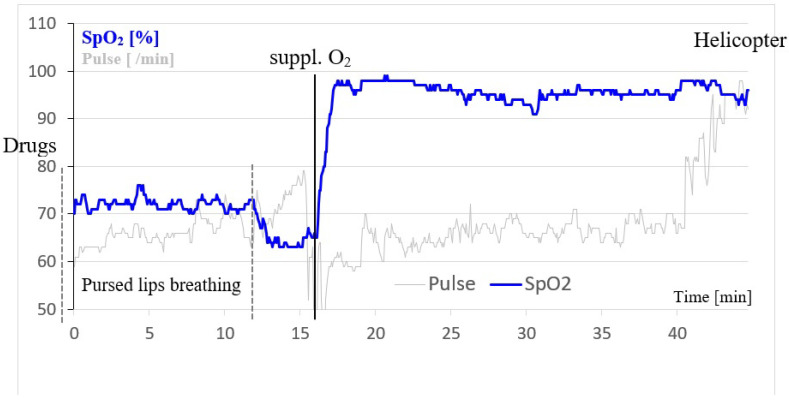
Time course of peripheral oxygen saturation (SpO_2_; blue) and pulse (grey) in O_2_-Pat at 4870 m. The black line indicates the start of supplemental oxygen at an initial flow rate of 3.0 L/min which was reduced to 0.5–1.0 L/min after several minutes. The grey dotted lines show the attempt of pursed lips breathing that had to be stopped after 12 min because of exhaustion. Drugs: administration of 8 mg of Dexamethasone, 20 mg of Nifedipine retard, and 800 mg of Ibuprofen. After 40 min O_2_-Pat walked to the helicopter (increasing pulse) using supplemental oxygen.

In AP-Pat SpO_2_ increased immediately after starting Auto-PEEP from <65% to ~80% with a further slow increase up to 95% at the end of the Auto-PEEP period (Figure 2). Interestingly, SpO_2_ was substantially higher and the heart rate substantial lower even 2 h after finishing Auto-PEEP compared with initial values. The steep oxygen binding curve in this region corresponds to an estimated increase in arterial pO_2_ of about 10 mmHg [15].

AP-Pat recovered overnight in a way that he was able to get up and walk around in the area of ABC with ascending to 4850 m, his LLS decreased to 6. No further Auto-PEEP was performed and the next day AP-Pat was able to climb the headwall to the ridge of the West Buttress (5050 m). Four days after Auto-PEEP he climbed with his wife to the summit of Denali (6190 m), starting from ABC (4330 m), and returned to it within one day without setting up high camp at 5200 m.

Both patients showed a distinct increase in SpO_2_ immediately after the start of therapy. The initial increase up to SpO_2_-levels of ~80% is similar in both patients within a few minutes with a constant fast increase in O_2_-Pat but subsequently slower increase in AP-Pat up to values close to 95% in both patients. What is remarkable is the slow decrease in SpO_2_ in AP-Pat after finishing Auto-PEEP.

## 4. Discussion

The similar increase in SpO_2_ in both patients is distinctive but both patients differ in several aspects.

The altitude at the onset of HAPE is higher in O_2_-Pat (5450 m vs. 4330 m). In general, one would expect a higher risk/severity of HAPE at a higher altitude. However, the impact of a delta of 1120 m relativizes for several reasons. First the impact of 5450 m in O_2_-Pat was only for a short time as he solely crossed Mera La Pass and descended down to Karre at 4870 m to his accommodation.

Second, there is a different density of the atmosphere, the “physiological” altitude near the arctic circle (63° north latitude) compared to Denali which is higher than in the Everest region (28° north latitude) [16,17,18]. At the altitude of the ABC at Denali, barometric pressure is 440 ± 2 mmHg [19], which is equivalent to an altitude of 4600 near Everest [17]. This results in a difference of their “physiological” altitude of 270 m.

Third, the altitude profiles of both patients are different. It took 9 days for O_2_-Pat to reach the altitude where he developed HAPE versus 4 days for AP-Pat. It is well-known that the risk of developing HAPE increases if the rate of ascent is fast resulting in an impaired acclimatization process [20]. With regard to this aspect the acclimatization profile of O_2_-Pat should have been better and more protective with regard to the risk of HAPE. In addition, AP-Pat had to carry heavy loads as at Denali a lot of equipment including fuel and nutrition for the whole expedition must be carried by the mountaineers themselves. AP-Pat had to carry more than 50 kg on a slide and in his rucksack. Heavy exercise is also a known risk factor for developing HAPE [21,22,23].

Comparing the two patients and their overall risk profiles, including their fitness we can state that they are similar.

Considerable difference exists regarding their age. There is a difference of 22 years that may result in different recovery abilities and even though younger persons are more prone to develop HAPE, there is evidence that systolic pulmonary arterial pressure increases more with age [24,25,26]. For AMS this is similar as the incidence and the severity of AMS are greater in younger individuals [27,28,29].

The most significant difference is the initial therapy and the onset of this treatment. The descent of O_2_-Pat is in line with the general recommendations for the therapy of high-altitude illness and a descent of nearly 600 m of altitude is within these recommendations [17]. As a result, O_2_-Pat recovered initially but symptoms worsened again. Probably the 600 m of descent were not enough [17] in his case and resulted in the need of further treatment. Self-medication with inhaled sympathomimetics and steroids is not the therapy of choice of HAPE and not recommended at SpO_2_-levels of 50% and below, even if it is questionable if these values are valid due to the complexity of SpO_2_ measurement at high altitude [30]. To sum, an efficient therapy did not start until the two physicians were involved, resulting in a delay of approximately 12 h. They measured a SpO_2_ of 62%, which is substantially higher than that measured by the patient. This value is similar to that of AP-Pat (58–64%). The treatment with Dexamethasone and Nifedipine is effective for HAPE [6,17], Ibuprofen is effective solely for headaches but not for HAPE. It is unlikely that the orally applied medications had an effect on the presented SpO_2_-curve (Figure 1) until the beginning of the evacuation as according to the drug information of the pharmaceutical company oral bioavailability with achievement of maximum serum levels is for Dexamethasone 60–120 min [31] and for Nifedipine 30–85 min [32], for retarded Nifedipine (as used in this case) it should be even longer.

In O_2_-Pat PLB had a substantially lower effect than Auto-PEEP in AP-Pat. Our assumption for this finding is that O_2_-Pat was not able to perform Auto-PEEP correctly because he was not able to follow instructions properly. Furthermore, he was too exhausted for this demanding procedure as is indicated by his increasing pulse. Probably the delayed onset of Auto-PEEP therapy due to the long-lasting mountain sickness had weakened O_2_-Pat so much that he did not have the strength for the demanding Auto-PEEP maneuver. One could speculate that there was already too great a diffusion barrier in his lungs due to HAPE, and Auto-PEEP could therefore not have the same effect as at the beginning of the disease.

As demonstrated by Figure 1, the application of bottled oxygen is a very effective therapy of HAPE regarding SpO_2_ or more precisely, due to difficult availability, a recommended way to bridge the time until descent [17].

By contrast, the therapy for AP-Pat started immediately after HAPE was diagnosed. A descent down to the last camp at the night was impossible because of difficult and dangerous terrain. As AP-Pat recovered during Auto-Peep-therapy no additional medication was necessary although Nifedipine as well as Dexamethasone were both available.

There are some anecdotal reports of PLB being effective in the therapy of HAPE even at the same location at Denali [33,34] but none of them had a similar long-lasting effect over 2 h as we found in AP-Pat. However, their PEEP was substantially lower by about 10 cm H_2_O. It is of interest in this context is that the accompanying three other healthy mountaineers showed a similar increase in SpO_2_ during Auto-PEEP apart from the fact that they started from a higher SpO_2_ level (median 78%). The striking difference was that their SpO_2_ decreased to initial levels after having finished Auto-PEEP making it likely that the high Auto-PEEP pressure probably leads to a structural improvement of oxygen transport into the bloodstream, most likely a reduction in the intra-alveolar fluids of the pulmonary edema and a persistent recruitment of atelectasis and thereby reduced hypoxic pulmonary vasoconstriction. We discussed this in more detail in our previous publication [10]. Other studies reported impressive improvement in HAPE or AMS patients using non-invasive continuous positive airway pressure (CPAP) or even bilevel positive airway pressure ventilation (BiPAP) with respiratory mask [35,36]. This confirms the good efficacy of increasing airway pressure in the therapy of HAPE, but again requires specialized equipment; therefore, it has the same limitations in its use in remote areas as bottled oxygen and hyperbaric chambers have.

Figure 3, modified from West et al. [37], Luks et al. [17] showed how altitude-induced hypoxia induces HAPE. With increasing altitude barometric pressure and with this partial oxygen pressure decrease, resulting in hypoxia which leads to an increase in PAP via the Euler-Liljestrand reflex (hypoxic pulmonary vasoconstriction) [38]. However, the vasoconstriction is uneven and may exceed the critical systolic pulmonary arterial pressure for the development of HAPE especially in HAPE-susceptible individuals [39]. This excessive pressure leads to stress failure with the development of pulmonary edema [17]. As a result, hypoxia is further exacerbated, creating a vicious circle.

Although Auto-PEEP breathing is strenuous, AP-Pat obviously benefited from the improved oxygenation because his pulse decreased during Auto-PEEP breathing and remained lower than the baseline values (see Figure 2). This is congruent with the study of Schoene et al. who described a constant heart rate with a tendency to decrease in HAPE patients under PEEP breathing in addition to the increase in SpO_2_ [34]. By contrast, healthy subjects’ SpO_2_ as well as heart rate increased under PEEP breathing. The authors conclude that HAPE patients benefit disproportionately well from improved oxygenation with PEEP breathing, but that this positive effect is offset in healthy subjects by the increased effort of breathing. Thus, healthy climbers are more likely to benefit from hyperventilation than from PEEP breathing [34].

Since Auto-PEEP breathing is easy to apply, based on our results it is recommended to have patients suffering from AMS or HAPE perform Auto-PEEP breathing every 2 h for 30 min as soon as severe AMS or HAPE is suspected. Accurate temporal monitoring of respiratory rhythm (2 s inspiration, 8 s expiration) is useful to avoid hyperventilation. Pulse-oximetric monitoring of these sick individuals as well as control of the therapy success on the basis of SpO_2_ and breathing difficulties is recommended. The pulse rate displayed simultaneously by the pulse oximeter can be used as an indicator of possible overexertion and thus a limited benefit of Auto-PEEP therapy [34]. If it increases during Auto-PEEP breathing, the expiration pressure should be reduced first; for example, to reduce the distance up to which the airflow of the exhalation can just be felt on the backside of the hand up to 20 cm. If pulse elevation persists, especially if there is a feeling of exertion, Auto-PEEP breathing should be stopped.

## 5. Limitations of the Study

The most important limitation of this study is the comparison between two individuals and the fact that the respective situations are similar but not identical. Therefore, no general conclusions can be drawn from this case report.

In remote areas the diagnosis of HAPE is based on the clinical finding with the cardinal symptom of a sudden significant loss in physical performance. In addition, dyspnea, a high LLS and low SpO_2_ can often be observed but none of them is 100% evidential for HAPE. Both patients showed these symptoms and the typical anamnesis, the altitude profile, and the course of the disease make it most likely that both suffered from HAPE. However, even if they suffered (only) from severe AMS, which is documented by their LLS-Score, the treatment (except Nifedipine in the O_2_-Pat) would have been identical.

The observation intervals of SpO_2_ (Figure 1 and Figure 2) differ, 45 min in O_2_-Pat and 205 min in AP-Pat. Therefore, we cannot make any statement about the sustainability of an increased SpO_2_ in O_2_-Pat after finishing supplemental O_2_ in the helicopter as it was observed in AP-Pat over two additional hours. Experience from other patients shows that there is no long-lasting effect of oxygen application.

## 6. Conclusions

An immediate onset of sufficient therapy is crucial if HAPE is suspected. The primary focus of all therapeutic measures is to improve oxygenation. However, this is often limited by organizational (difficulty of descent) or logistic reasons (availability of hyperbaric chambers and bottled oxygen) and therefore leads to considerable delays in the start of the therapy. Auto-PEEP ensures an immediate start of therapy with an immediate increase in SpO_2_ and shows promising improvement of the clinical situation in the presented case. We recommend 30 min of Auto-PEEP every 2 h until recovery or availability of established therapy options of HAPE.

## Figures and Tables

**Figure 2 ijerph-19-16185-f002:**
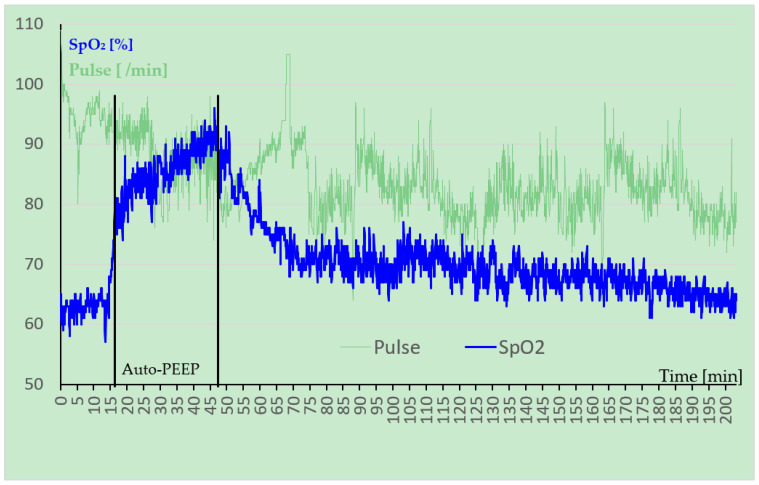
Time course of SpO_2_ (blue) and pulse (green) in AP-Pat at 4380 m. The black lines indicate beginning and end of Auto-PEEP breathing.

**Figure 3 ijerph-19-16185-f003:**
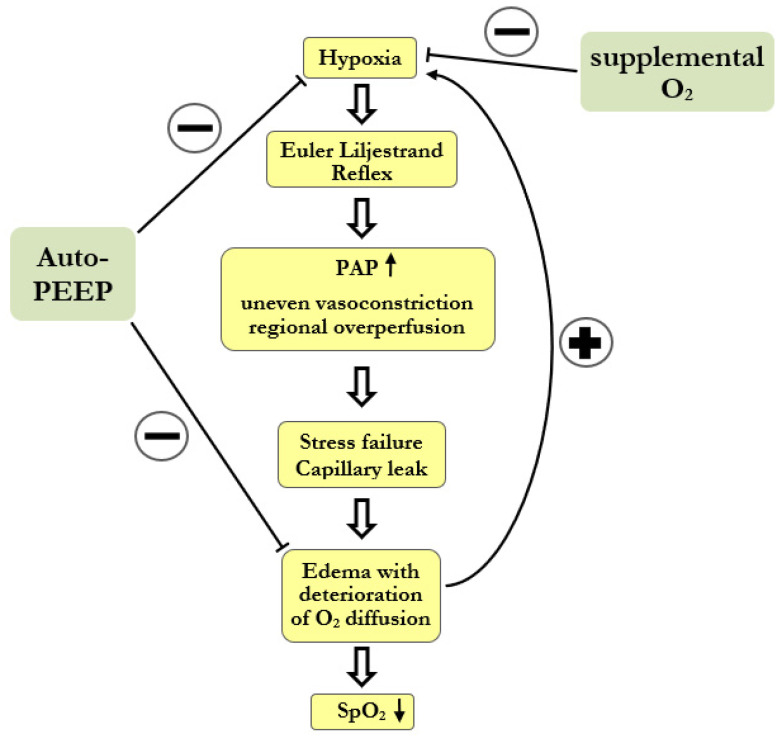
**How hypoxia induces HAPE, modified 
from West et al.** [37] considering Luks et 
al. [17] with the points of attack Figure 2. The pulmonal edema aggravates hypoxia due to enlarged alveolar-capillary diffusion distance. 
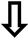
: leads to; 
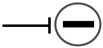
: reduces;
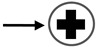
: increases.

## Data Availability

The data presented in this study are available on request from the corresponding author.

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
