# Peer review of "Initial Treatment of High-Altitude Pulmonary Edema: Comparison of Oxygen and Auto-PEEP"

_ijerph, 2022, doi:10.3390/ijerph192316185_

Round 1
Reviewer 1 Report
Dear authors,
thank you for the opportunity to review this extremely interesting and relevant paper. The chance to have an easy on-scene therapy available wherever the person may be at the momentwhen the potential life-threatening HAPE has been diagnosed is grest. This paper should be pzublished at all costs!
Principally it may be published as it is, but please consider a few suggestions:
Line 37: I am not a native speaker, but please check whether it is correct as it is: "expectations SET in them" or would it be correct to say "expectations PUT in them"?
Line 49: Your reference Bärtsch 1993 is an important paper but nowadays not the Gold Stanmdard anymore. You may rerefence it but I would suggest to add the actual Gold Standards as there is the recommendation of the international climbing federation (www.theuiaa,org/medical_advice.html, recommendations No.2 and 3). You may also reference the WMS recommendation.
Lines 57-64: very good description for anybody who may use it for the first time!
Lines 102-105, 174-177: I suggest to add the antithetical results published by Hundt et al. (Hundt, N.; Apel, C.; Bertsch, D.; van der Giet, M.S.; van der Giet, S.; Grass, M.; Gschwandtl, C.; Heussen, N.; Kühn, C.; Müller-Ost, M.; Risse, J.; Schmitz, S.; Timmermann, L.; Wernitz, K.; Gieseler, U.; Küpper, T.Factors influencing pressure and volume of the pulmonary circulation as risk factors for developing a High Altitude Pulmonary Edema. IJERPH, in press). They have checked mountaineers who ascended Kalar Patthar with different loads and speed, have got an echocardiography immediately when they arrived at the summit (<1' after summiting) and there was no correlation between exercise and pulmonary pressure.
Line 284: Because auto-PEEP is so easy, everywhere available and may be used without any delay at any mountain worldwide you may suggest in your conclusions that UIAA may include it as Frst Aid option in the next update of their recommendation.
Again: Congratulations! Go ahead!
Author Response
Reviewer 1.
Dear Reviewer:
Thank you very much for your helpful comments. Please, find our responses below. All changes in the revised manuscript are highlighted (yellow).
Dear authors,
thank you for the opportunity to review this extremely interesting and relevant paper. The chance to have an easy on-scene therapy available wherever the person may be at the momentwhen the potential life-threatening HAPE has been diagnosed is grest. This paper should be pzublished at all costs!
Principally it may be published as it is, but please consider a few suggestions:
Line 37: I am not a native speaker, but please check whether it is correct as it is: "expectations SET in them" or would it be correct to say "expectations PUT in them"?
Re: Thank you for proofreading. We have replaced “set” by “put”
Line 49: Your reference Bärtsch 1993 is an important paper but nowadays not the Gold Stanmdard anymore. You may rerefence it but I would suggest to add the actual Gold Standards as there is the recommendation of the international climbing federation (www.theuiaa,org/medical_advice.html, recommendations No.2 and 3). You may also reference the WMS recommendation.
Re: Thank you for this valuable suggestion. We added the following reference in Line 49
Angelini, C.; Basnyat, B.; Bogg, J.; Chioconi, A. R.; Domej, W.; Ferrandis, S.; Gieseler, U.; Hefti, U.; Hillebrandt, D.; Holmgren, J.; Horii, M.; Jean, D.; Koukoutsi, A.; Kubalova, J.; Küpper, T.; Meijer, H.; Milledge, J.; Morrison, A.; Mosaedian, H.; Omori, S.; Rotman, I.; Schoeffl, V.; Shahbazi, J.; Windsor, J. CONSENSUS STATEMENT OF THE UIAA MEDICAL COMMISSION VOL: 2 Emergency Field Management of Acute Mountain Sickness, High Altitude Pulmonary Edema, and High Altitude Cerebral Edema. (25.11.2022),
as well as the 2019 update of the Wilderness Medical Society:Luks, A. M.; Auerbach, P. S.; Freer, L.; Grissom, C. K.; Keyes, L. E.; McIntosh, S. E.; Rodway, G. W.; Schoene, R. B.; Zafren, K.; Hackett, P. H., Wilderness Medical Society Clinical Practice Guidelines for the Prevention and Treatment of Acute Altitude Illness: 2019 Update. Wilderness Environ Med 2019, 30, (4S), S3-S18.
Lines 57-64: very good description for anybody who may use it for the first time!
Re: Thank you very much
Lines 102-105, 174-177: I suggest to add the antithetical results published by Hundt et al. (Hundt, N.; Apel, C.; Bertsch, D.; van der Giet, M.S.; van der Giet, S.; Grass, M.; Gschwandtl, C.; Heussen, N.; Kühn, C.; Müller-Ost, M.; Risse, J.; Schmitz, S.; Timmermann, L.; Wernitz, K.; Gieseler, U.; Küpper, T.Factors influencing pressure and volume of the pulmonary circulation as risk factors for developing a High Altitude Pulmonary Edema. IJERPH, in press). They have checked mountaineers who ascended Kalar Patthar with
different loads and speed, have got an echocardiography immediately when they arrived at the summit (<1' after summiting) and there was no correlation between exercise and pulmonary pressure.
Re: This interesting paper, seems to contradict the established opinion of additional weight load on HAPE risk. [1-3]. But in the mentioned paper the weight load of backpacks is not discussed in detail, it is simply mentioned that there was no statistical correlation without providing detailed data (p value, mean/median etc.) In addition, the time point of measurement (< 1 min) is not mentioned in the text (by the way, less than one minute seems very short for undressing several layers of clothing, entering a tent, placing the patient, starting measurement and getting valid results.). Furthermore, exertion is not only dependent from backpack load but also from physical fitness and ascent speed. A very fit person carrying for example 20 kg will not be as exhausted as an unfit person carrying the same weight. Finally, no people suffering from HAPE were examined. We therefore do not feel, that this article does really proof that weight load has no influence on HAPE risk. Physical exercise does always rise systemic and pulmonal vascular resistance and therefore blood pressure will rise in the systemic and as well in the pulmonary circulation. The values obtained in the mentioned study were values after exercise, most likely also after initial recovery. In susceptible or even ill persons different sPAP values are likely to our opinion.
However, we have already cited this paper in our manuscript (former Lit No 22, now No 24) with regard of influence of age on sPAP changes. Here a sound statistical analysis is given in the recommended article. We have added “there is evidence that systolic pulmonary arterial pressure will increase more with age”
Line 284: Because auto-PEEP is so easy, everywhere available and may be used without any delay at any mountain worldwide you may suggest in your conclusions that UIAA may include it as Frst Aid option in the next update of their recommendation.
Re: We appreciate the UIAA and further alpine rescue associations like ICAR and WMS very much and are in contact with some of the members. However, we do not think that it is appropriate to add such a demand in our paper. The beforementioned commissions do regular literature research, so this article will be part of updates.
Again: Congratulations! Go ahead!
Re: Thank you
Reviewer 2 Report
Reviewer’s comments:
Increase of SpO2 above 80% in the therapy of high-altitude pulmonary edema (HAPE) due to PAP by using AP-Pat method after starting Auto-PEEP was exposed in the case study to improve the disadvantages of hyperbaric chamber or the use of bottled oxygen methods.
In section “3. Results”, the time sequence of things in this case study was suggested by using a flow chart diagram.
In section “4. Discussion”, the Inter-relationship diagraph should be presented to clearly exhibit the disease causality including therapy processes.
Author Response
Reviewer 2:
Dear Reviewer:
Thank you very much for your helpful comments. Please, find our responses below. All changes in the revised manuscript are highlighted (yellow).
Increase of SpO2 above 80% in the therapy of high-altitude pulmonary edema (HAPE) due to PAP by using AP-Pat method after starting Auto-PEEP was exposed in the case study to improve the disadvantages of hyperbaric chamber or the use of bottled oxygen methods.
Re: Yes, because often bottled oxygen or a hyperbaric chamber are not available at the site of the disease. Auto-PEEP can be performed anywhere if the person is instructed in this procedure before at good health (e.g. high altitude courses, base camp …) and is still able to perform it.
In section “3. Results”, the time sequence of things in this case study was suggested by using a flow chart diagram.
Re: Thank you very much for this suggestion. To clarify the sequence of different therapies we added the following additional information to fig 1: “drugs” “PLB”, “helicopter”. The timeline can easily be taken from the x-axis.
In section “4. Discussion”, the Inter-relationship diagraph should be presented to clearly exhibit the disease causality including therapy processes.
Re: Thank you very much for this great suggestion. We have added an inter-relationship diagraph (fig 3) in the discussion section and explained it in the manuscript as follows:
Figure 3, modified from West et al [4] considering Luks et al [5] shows how altitude-induced hypoxia induces HAPE. With increasing altitude barometric pressure and with this partial oxygen pressure decrease, resulting in hypoxia which leads to an increase in PAP via the Euler-Liljestrand reflex (hypoxic pulmonary vasoconstriction). [6] But the vasoconstriction is uneven and may exceed the critical systolic pulmonary arterial pressure for the development of HAPE especially in HAPE-susceptible individuals. [7] This excessive pressure leads to stress failure with the development of pulmonary edema. [5] As a result, hypoxia is further exacerbated, creating a vicious circle.
Thank you again for the helpful suggestions.